# Intersecting distributed networks support convergent linguistic functioning across different languages in bilinguals

Shujie Geng[1,2], Wanwan Guo[1,2], Edmund T. Rolls [1,3,4], Kunyu Xu[5], Tianye Jia [1,2], Wei Zhou[6], Colin Blakemore[7], Li-Hai Tan [8,9✉], Miao Cao [1,2✉] & Jianfeng Feng [1,2✉]

How bilingual brains accomplish the processing of more than one language has been widely investigated by neuroimaging studies. The assimilation-accommodation hypothesis holds that both the same brain neural networks supporting the native language and additional new neural networks are utilized to implement second language processing. However, whether and how this hypothesis applies at the finer-grained levels of both brain anatomical organization and linguistic functions remains unknown. To address this issue, we scanned Chinese-English bilinguals during an implicit reading task involving Chinese words, English words and Chinese pinyin. We observed broad brain cortical regions wherein interdigitated distributed neural populations supported the same cognitive components of different languages. Although spatially separate, regions including the opercular and triangular parts of the inferior frontal gyrus, temporal pole, superior and middle temporal gyrus, precentral gyrus and supplementary motor areas were found to perform the same linguistic functions across languages, indicating regional-level functional assimilation supported by voxel-wise anatomical accommodation. Taken together, the findings not only verify the functional independence of neural representations of different languages, but show co-representation organization of both languages in most language regions, revealing linguistic-feature specific accommodation and assimilation between first and second languages.

[1] Institute of Science and Technology for Brain-Inspired Intelligence, Fudan University, Shanghai 200433, China. [2] Key Laboratory of Computational Neuroscience and Brain-Inspired Intelligence (Fudan University), Ministry of Education, Shanghai 200433, China. [3] Department of Computer Science, University of Warwick, Coventry CV4 7AL, UK. [4] Oxford Centre for Computational Neuroscience, Oxford, UK. [5] Institute of Modern Languages and Linguistics, Fudan University, Shanghai 200433, China. [6] Beijing Key Laboratory of Learning and Cognition, School of Psychology, Capital Normal University, Beijing 100037, China. [7] Department of Neuroscience, City University of Hong Kong, Hong Kong 999077, China. [8] Guangdong-Hongkong-Macau Institute of CNS Regeneration and Ministry of Education CNS Regeneration Collaborative Joint Laboratory, Jinan University, Guangzhou 510632, China. [9] Center for Language and Brain, Shenzhen Institute of Neuroscience, Shenzhen 518057, China. ✉email: tanlh@sions.cn; mcao@fudan.edu.cn; jianfeng64@gmail.com

Bilingual individuals exhibit remarkable competence in processing more than one language. However, it remains open to what extent this relates to common neural systems for different languages, or whether different neural systems become specialized for each language. A congenital determinism view derived from Noam Chomsky assumes that a language mechanism in the human brain is the prerequisite, and that different language systems are fitted to the same system by adjusting parameters. In line with this view, evidence has been described that the first language (L1) and second language (L2) activate the same brain regions such as the left inferior frontal gyrus (IFG)[1,2]. In contrast, a growing number of studies found that separate brain mechanisms exist in bilinguals' brain, potentially related to different environmental factors such as language types[3], acquisition order[4], and age at acquisition[5]. To resolve the paradox, Perfetti and colleagues proposed the assimilation-accommodation hypothesis[6]. This states that the human brain not only utilizes neural networks supporting the native language to implement second language processing (known as assimilation), but also recruits new neural networks to adapt to unique linguistic features of the second language (known as accommodation). In line with the assimilation-accommodation framework, studies have reported that the degree of similarity between the neural networks of language 1 (L1) and L2 is affected by the distance between L1 and L2, acquisition age for L2, and proficiency of L2[7–9].

However, what the general organization principles are for the neural changes for the assimilation-accommodation for L1 and L2 remain unclear. Our previous work employed the multivariate analysis method to depict the distinct distributed patterns of neural activity for L1 and L2[10]. This demonstrated the important possibility that the two languages involve anatomically interleaved but functionally independent neural populations within a given cortical region, and thus, distinct patterns of neural computations are pivotal for bilingual speakers to appropriately use each language. Notably, the fundamental logic of language preprocessing may remain the same across different languages. For word recognition, three language-processing representational components, orthographic, phonological, and semantic

representations, were found to be processed and integrated in brain language networks during reading to integrate information from visual properties and pronunciations of printed words and to access semantics[11–14]. In this context, the detailed organization rules for the two languages in bilinguals across the whole language network to accomplish the same linguistic functions (i.e., orthographic, phonological and semantic) based on anatomically identifiable regions need to be further revealed.

To address these gaps, we utilized representational similarity analysis (RSA), which is powerful for integrating different scale activities to identify cognitive manipulation[12,15–18], with the neuroimaging data from a group of Chinese-English bilingual individuals ($n = 51$) during performance of an implicit reading task for Chinese words, English words, and Chinese pinyin (Fig. 1a). Specifically, English is an alphabetic language whereas Chinese is a logographic language[19], and they differ significantly in their visual-spatial properties, orthographic rules (i.e., the regularity of mapping from graphemes to phonemes) and semantic access[20–22]. Meanwhile, Chinese pinyin, as a phonetic symbol system for Chinese characters, shares similar alphabetic orthography-phonology mapping properties with English words but uses the same phonology and semantic lexicons as Chinese words, the underlying neural mechanisms for which may be partially similar for Chinese words and English words[23]. All participants had received a unified education of Chinese pinyin at the elementary level in mainland China and passed the College English Test Band 4. We hypothesized that (1) interleaved neural populations responding to the same linguistic components across languages would be found, which reflects the accommodation of L2; (2) meanwhile, the same linguistic components would be processed within the same regions across languages, which reflects the assimilation of L2 based on L1.

## Results

**Behavioral performance and brain activations across different languages.** The accuracy and reaction time results for the participants are shown in Fig. 1b. Group mean accuracy of Chinese words, English words and Chinese pinyin reading were 98.4% (87.5–100%, standard deviation (SD) = 2.4%), 98.1% (80–100%,

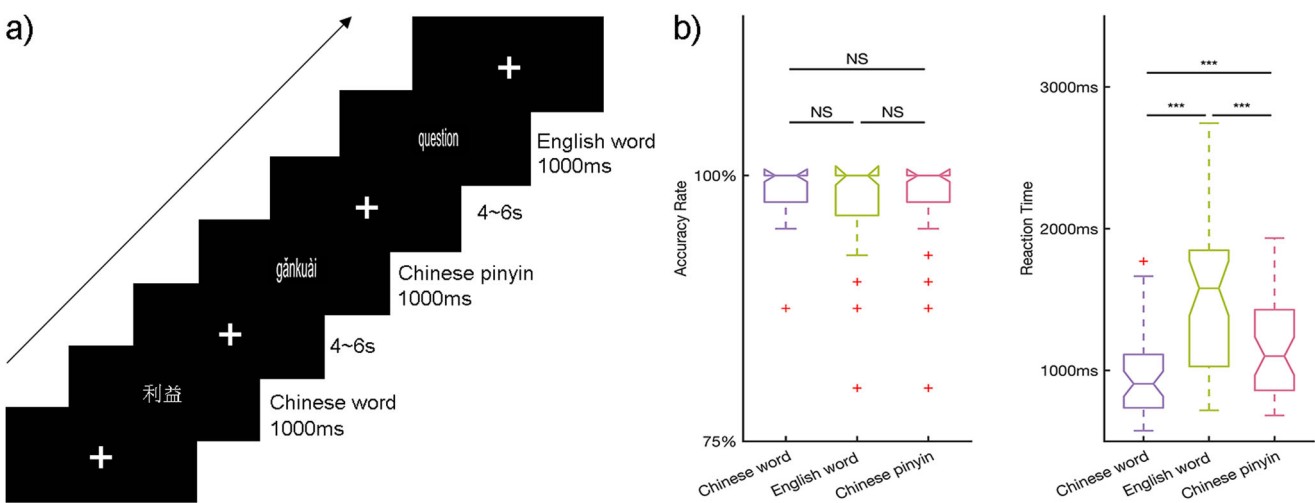

**Fig. 1 Experimental design and task performance. a** An event-related design was utilized in the current study. Visual stimuli of Chinese word, English word, and Chinese pinyin reading tasks were presented for 1000 ms in randomized order with an inter-stimulus interval of 4–6 s. Responses by participants were required to identify language types after semantic access. **b** Accuracy rate and reaction time for Chinese word, English word, and Chinese pinyin reading tasks. Reaction time was counted from the onset of visual stimuli to the button-press response. A one-way ANOVA was conducted to test significant differences between language types. No significant differences were found between accuracy rates due to the ceiling effect but significant differences were found between reaction times. Post hoc analysis (Bonferroni corrected, $p < 0.05$) revealed the longest reaction time for accessing the semantics of Chinese pinyin and the shortest time for Chinese words.

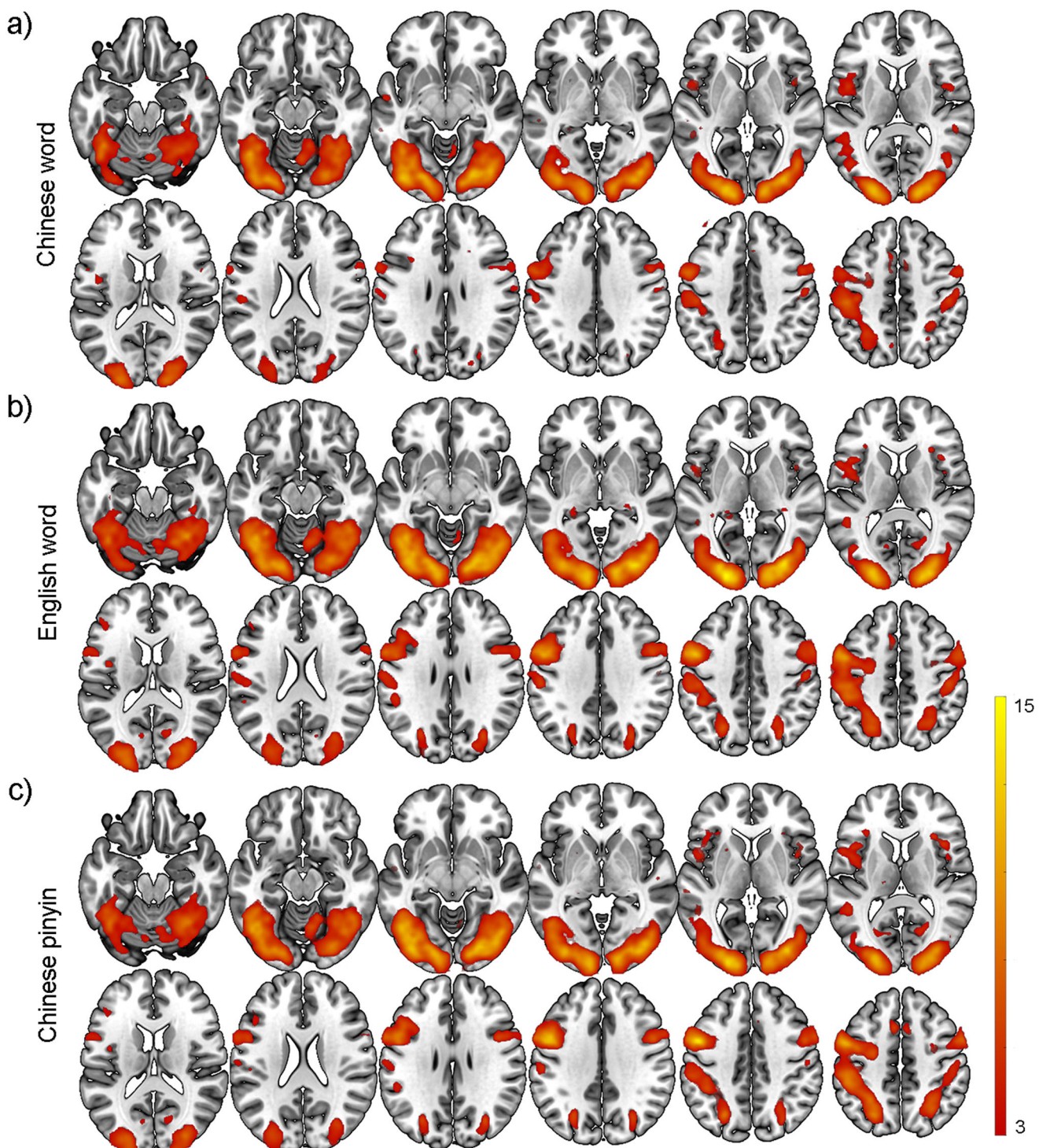

**Fig. 2 Activation analysis for Chinese words, English words, and Chinese pinyin reading tasks.** Second-level general linear models (GLMs) were performed separately to obtain activation maps for the recognition of **a** Chinese words, **b** English words, and **c** Chinese pinyin. Brighter colors indicate higher t values. (Voxel-wise $p < 0.05$, FDR corrected $p < 0.05$, and cluster size >10).

SD = 4.0%), and 97.16% (80–100%, SD = 5.0%), respectively, and there was no significant difference across conditions ($F$ (2, 86) = 1.20, $p = 0.31$). The group mean reaction time for Chinese words, English words and Chinese pinyin reading were 968 ms (574–1770 ms, SD = 293 ms), 1182 ms (682–1933 ms, SD = 359 ms), 1497 ms (718–2744 ms, SD = 518 ms), respectively. Significant differences between conditions were revealed by one-way ANOVA ($F$ (2, 86) = 19.3, $p < 0.01$) and post hoc two-sample t-test ($ps < 0.01$, Bonferroni correction) with increasing reaction time for reading Chinese words, English words and Chinese pinyin, used to indicate successful semantic access when pressing a button.

Typical brain activation maps at the group-level were calculated for each reading condition (Fig. 2, $p < 0.05$ with false discovery rate (FDR) correction, cluster size >10 voxels). Highly similar patterns in the activation maps across the three languages

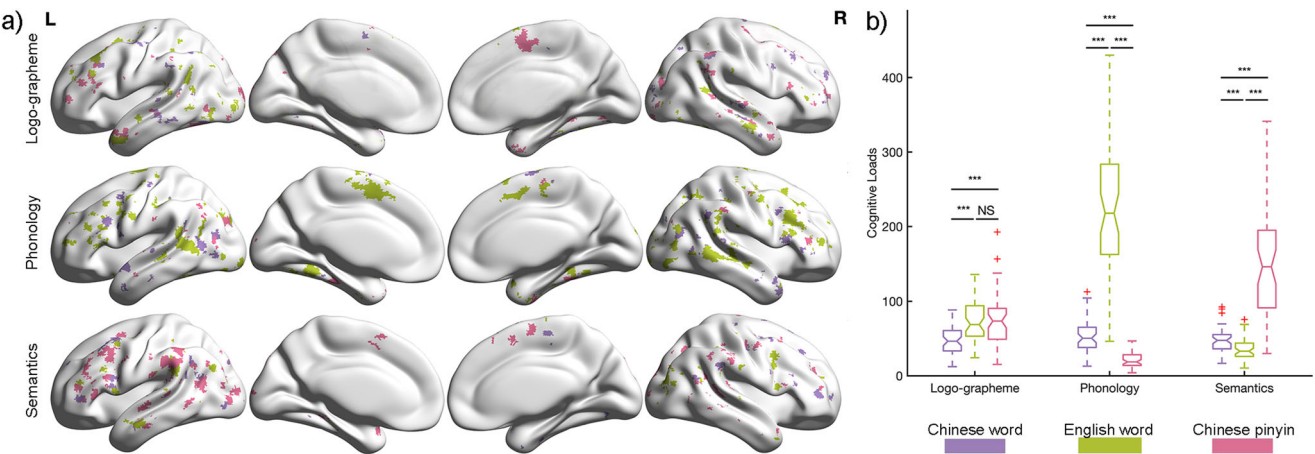

**Fig. 3 Brain responses to linguistic component for three language types. a** Brain involvement maps of 3 reading systems for each linguistic component. Purple indicates brain activations elicited by Chinese word, pistachio green represents English word and magenta denotes Chinese pinyin. Gray lines indicate the regional boundaries. **b** Comparisons in the cognitive loads between language types and linguistic components. Two-way repeated ANOVA was performed, and significant interactions were found between linguistic components and language types. Post hoc analysis shows differentiated patterns of brain activity elicited by language-processing components across language types. ***$p < 0.001$; NS, non-significance.

were observed (Chinese words vs. Chinese pinyin, $r = 0.90$, $p < 0.001$; Chinese words vs. English words, $r = 0.93$, $p < 0.001$; and Chinese pinyin vs. English words, $r = 0.97$, $p < 0.001$). Specifically, the middle temporal gyrus, fusiform areas, middle occipital gyrus, middle frontal gyrus, superior temporal gyrus, and inferior frontal gyrus were consistently activated under all conditions, in line with previous studies. Bonferroni corrected activation maps, as shown in Supplementary Fig. 1, indicated that the clusters remained and were thus validated.

**Accommodation for L2 in the divergent neural representation patterns for language-processing components**. We then utilized the RSA method to explore the brain responses to linguistic components during language processing in every language within a predefined language-related mask including the cortical regions found above and which were consistently reported to be involved in language comprehension-related processing as well as their symmetric regions in the right hemisphere. We first constructed three behavioral representational dissimilarity matrices (logo-grapheme RDMs, phonological RDMs, and semantic RDMs) based on word units, phonetic features and word2vec. Neural representational dissimilarity matrices (neural RDMs) were then constructed to depict the brain activity patterns for each writing system. Through examining the correspondence relationships between the behavioral RDMs and neural RDMs, we functionally localized voxels sensitive to each language-processing component. Finally, we obtained the significant correlation maps for all language-processing components, i.e., the representation maps, across writing systems for each participant with the significance threshold set as $p < 0.05$ with cluster size >10 voxels.

The brain representation maps of each language system for every linguistic component at the group level are shown in Fig. 3a. A separate pattern across languages for all three components was detected throughout all cortical areas within the language mask. Specifically, interdigitated distributed neural populations within each region were evoked for processing the same linguistic components across different languages. To evaluate the magnitude of the brain response, we calculated the cognitive loads for every linguistic component in each language for every subject, which were defined as the sum of the correlation values of all significant voxels. Comparisons of cognitive loads between language linguistic components and writing systems are

shown in Fig. 3b. For logo-grapheme processing, minimal loads were detected during Chinese word reading. The highest loads for phonology processing were English words, and the highest loads for semantic processing were Chinese Pinyin.

To further explore the degree of separation of neural responses for different linguistic components across languages, we conducted a classification analysis based on support vector regression (SVR) with leave-one-subject-out cross validation. We found that the category of ideographic and alphabetic writing system could be 100% correctly classified with the representation maps of all linguistic components as features (accuracy: 100% between Chinese words and English words, 100% between Chinese words and Chinese pinyin, 96.1% between English words and Chinese pinyin, Fig. 4a). Similar results were found by employing the brain representations of each single component as features (accuracy based on logo-grapheme: 100% between Chinese words and English words, 100% between Chinese words and Chinese pinyin, 82.4% between English words and Chinese pinyin; accuracy based on phonology: 100% between Chinese words and English words, 100% between Chinese words and Chinese pinyin, 96.1% between English words and Chinese pinyin; accuracy based on semantic: 100% between Chinese words and English words, 100% between Chinese words and Chinese pinyin, 100% between English words and Chinese pinyin, Fig. 4a). With the recursive feature elimination (RFE) scheme based on leave-one-sample-out cross validation, we ranked the brain regions in terms of their contributions to predict language types. The brain areas that contributed significantly are shown in Fig. 4b, with the brain activity elicited by logo-grapheme processing in the left middle frontal cortices contributing the most.

**Assimilation of L2 based on L1 in the convergent linguistic domains for language-processing components at the regional level**. While the above results revealed the separate response patterns in linguistic components across languages in corresponding to the accommodation for L2, we still searched for potential mechanisms of assimilation of L2 based on L1. To address this problem, we calculated the overlapping areas of neural representations for all components between any two languages (Fig. 5 and Supplementary Table 1). Shared brain involvement in core regions across writing systems were found. For logo-grapheme processing, overlapping brain representations

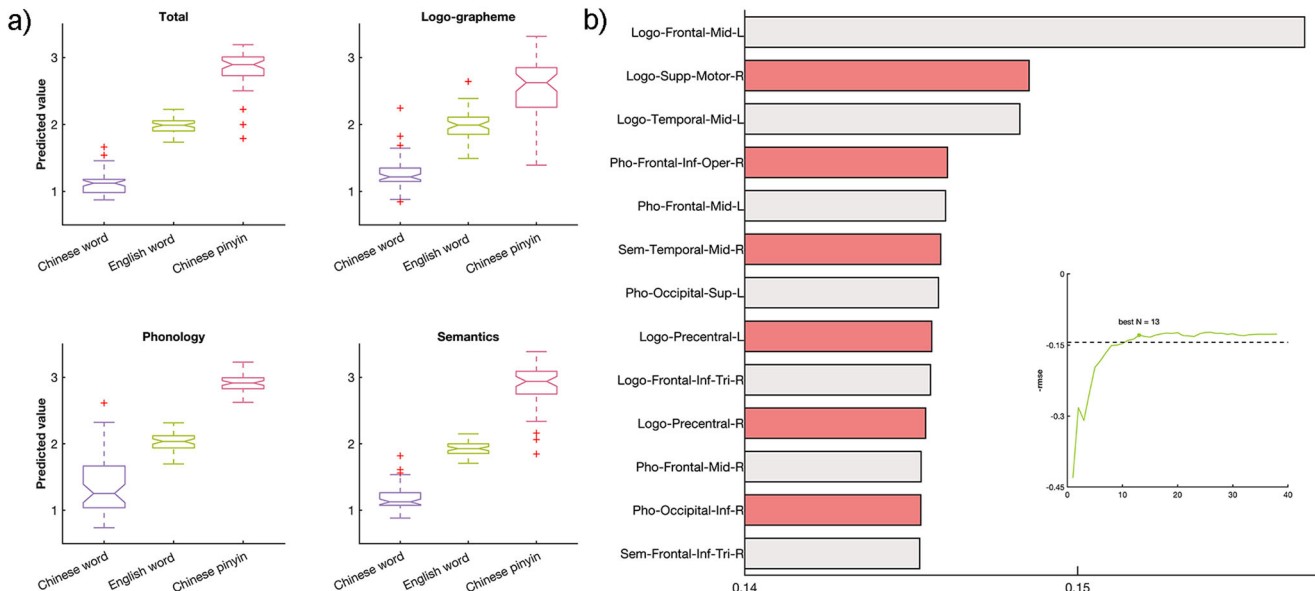

**Fig. 4 Separate brain activity patterns of linguistic components across language types revealed by the support vector regression model. a** The prediction results of language types with the brain activity elicited by language-processing components in 30 ROIs as input features. **b** The bars indicate the contributions of regions to the model obtained from a recursive feature elimination scheme. The plot indicates the best number of features for classification. Abbreviations, Logo Logo-grapheme, Pho Phonology, Sem Semantics.

were found in the right fusiform, right superior temporal cortex, left middle frontal cortex, and right inferior frontal cortex for reading Chinese words and English words; the right superior temporal cortex, right inferior frontal gyrus (rIFG), right inferior parietal cortex and right precentral gyrus for reading Chinese words and Chinese pinyin; and the left middle temporal cortex, left inferior occipital cortex, and right superior temporal cortex for reading Chinese pinyin and English words. For phonological involvement, overlapping brain activity was found in the right middle temporal cortex, right middle occipital cortex, left middle temporal cortex, left superior temporal cortex, left superior occipital cortex, and right dorsal inferior frontal cortex for reading Chinese words and English words; and the right fusiform, left superior occipital cortex, right supramarginal gyrus and right precentral gyrus for reading Chinese pinyin and English words. No overlap in brain activity was found between reading Chinese words and Chinese pinyin. For semantic processing, overlapping brain activity was found in the left precentral gyrus for reading Chinese words and English words; the left ventral inferior frontal gyrus, left middle temporal cortex, left angular cortex and right middle temporal cortex for reading Chinese words and Chinese pinyin; and the left ventral inferior frontal gyrus and left supramarginal gyrus for reading Chinese pinyin and English words. Notably, we found that the triangular parts of the left inferior frontal gyrus corresponding to semantic processing was consistent for all three languages, although the overlaps failed to reach the threshold of a cluster size larger than 10 voxels.

Next, we explored the similarity in involvement of the three linguistic components across languages for every language-related region. Based on the brain representation maps of all language-processing components (logo-grapheme, phonology, and semantics) for each language (Fig. 6a), we found that most brain regions were involved in two or three language-processing components. Specifically, the involvement in linguistic processing in a majority of language-related regions was universal for L1 and L2, including the opercular part of the left inferior frontal gyrus which was consistently involved in brain activity pertaining to phonological and semantic processing independently of language; and the left precentral gyrus, left superior temporal gyrus, left middle

temporal gyrus and left temporal pole which were consistently involved in brain processing of logo-grapheme, phonology and semantics independently of language (Fig. 6c). Meanwhile, some regions were only partially consistent in linguistic component involvement between the first and second language, which might be due to different demands of linguistic features (Fig. 6d). The left middle frontal gyrus, left angular gyrus and left inferior parietal gyrus showed activity during logo-grapheme and semantic processing in Chinese reading but logo-grapheme and phonological involvement in English; the triangular parts of the left inferior gyrus showed activity during phonological and semantic processing in Chinese reading but phonological, logo-grapheme and semantic processing in English reading. The left supramarginal gyrus demonstrated activity during phonology and logo-grapheme processing in Chinese reading but phonological and semantic processing in English reading. Chinese word reading showed equivalent cognitive loads for the three linguistic components (logo-grapheme, phonology, and semantics). Meanwhile, reading English words was associated with high cognitive loads for phonology, and Chinese pinyin for semantic processing. In addition, comparisons of cognitive loads between language linguistic components and writing systems showed that Chinese word reading exhibited equivalent cognitive loads for the three linguistic components (logo-grapheme, phonology, and semantics). Meanwhile, reading English words was associated with high cognitive loads for phonology, and Chinese pinyin for semantic processing (Fig. 6b).

**Significant association between cognitive loads and behavioral performance across languages.** Finally, to explore the potential effects of neural loads on behavioral performance, we conducted an association analysis between cognitive loads and reaction time with partial correlation. We found significant correlations between the reaction time for reading Chinese words and logo-grapheme brain loads ($r = 0.454$, $p < 0.05$, Bonferroni corrected), and reading Chinese pinyin and semantic brain loads ($r = 0.528$, $p < 0.01$, Bonferroni corrected), as shown in Fig. 7a–c. We also detected marginally significant correlations between the reaction

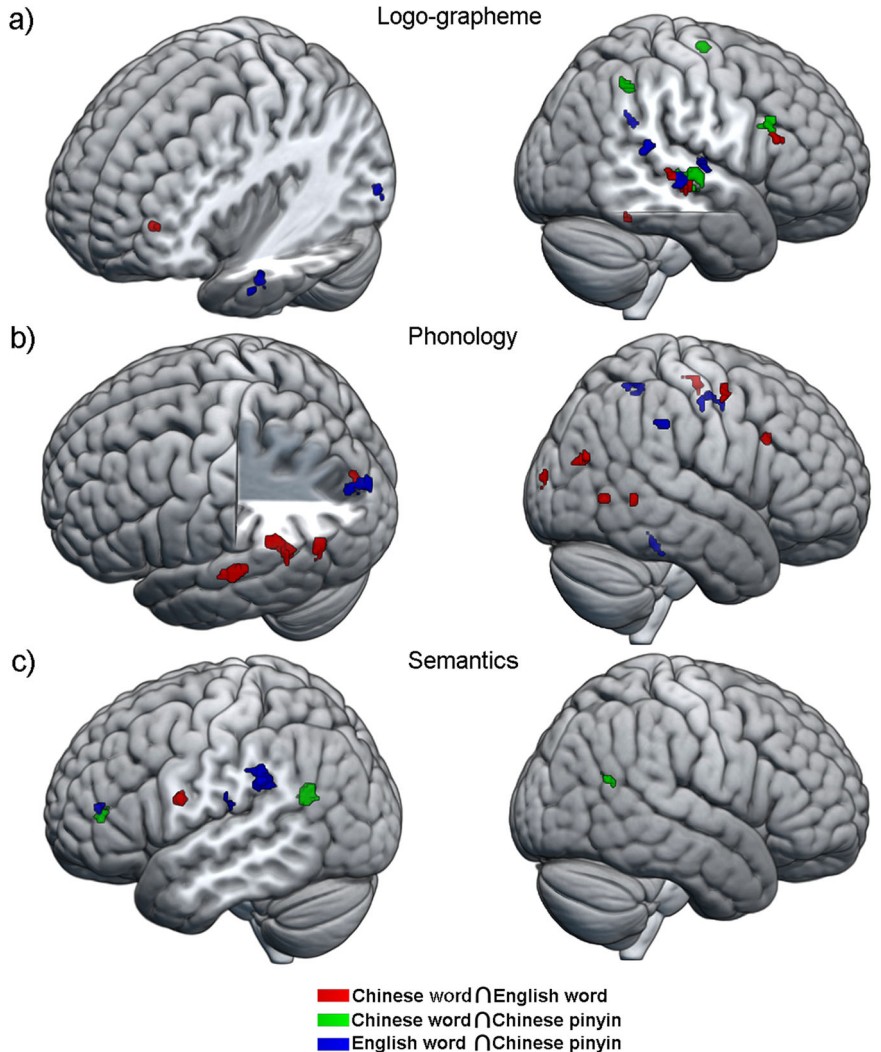

**Fig. 5 Overlap of brain involvement among language types in each linguistic component.** The overlapping patterns of brain load for **a** logo-grapheme, **b** phonology, and **c** semantics between pairs of languages. Red represents overlapping brain involvement between Chinese words and English words; green signifies overlapping brain activity between Chinese words and Chinese pinyin; and blue indicates an overlap in brain activity between English words and Chinese pinyin.

times for reading English words and phonological brain loads ($r = -0.309$, $p = 0.047$, uncorrected); and reading Chinese pinyin and logo-grapheme brain loads ($r = 0.303$, $p = 0.051$, uncorrected). Pearson's correlation analyses were also conducted between the total brain loads and reaction time for every language type, but no significant correlations were found.

**Results of validation analyses.** Given that age of acquisition (AOA) for a second language has potential effects on the distribution and involvement of neural networks supporting L2 and L1 in bilinguals, we divided participants into an early AOA subgroup (3–8 years) and a late AOA subgroup (9–15 years) and depicted brain load maps of linguistic components for early AOA and late AOA respectively. As shown in Fig. 8a, both early AOA and late AOA participants showed intersecting neural populations underlying linguistic components for Chinese words, English words, and Chinese pinyin. Specifically, early AOA participants showed similar brain load maps of linguistic components between Chinese words and English words. Compared with early AOA participants, late AOA participants activated more extensive brain regions processing L2 ($P_{permutation} =$

0.060, 10,000 times). Furthermore, at regional level, despite of AOA, significant interactions between language types and linguistic components were remain: no-significant differences of brain loads among linguistic components in Chinese words, reversed V shape in English words, and V shape in Chinese pinyin (Fig. 8b), denoting brain adaptations for languages.

Validation analyses were performed to check that the patterns of cortical involvement for different linguistic components in the different languages were replicable. We found that similar anatomically separated patterns of linguistic components were stable at the individual level. Also, reliable neural response patterns were found when randomly selecting subsamples and excluding non-right-handed subjects (Supplementary Figs. 2–3 and Supplementary Table 2). In addition, the relevant results were stable when the definition for cognitive loads was the summed number of significant voxels, instead of the summed number of correlation values (Supplementary Tables 3–4). Supplementary Fig. 4 shows the significant neural representations on HCPex template. The numbers of significant voxels across every region in the HCPex template[24] were also calculated and presented in Supplementary Data 1–3.

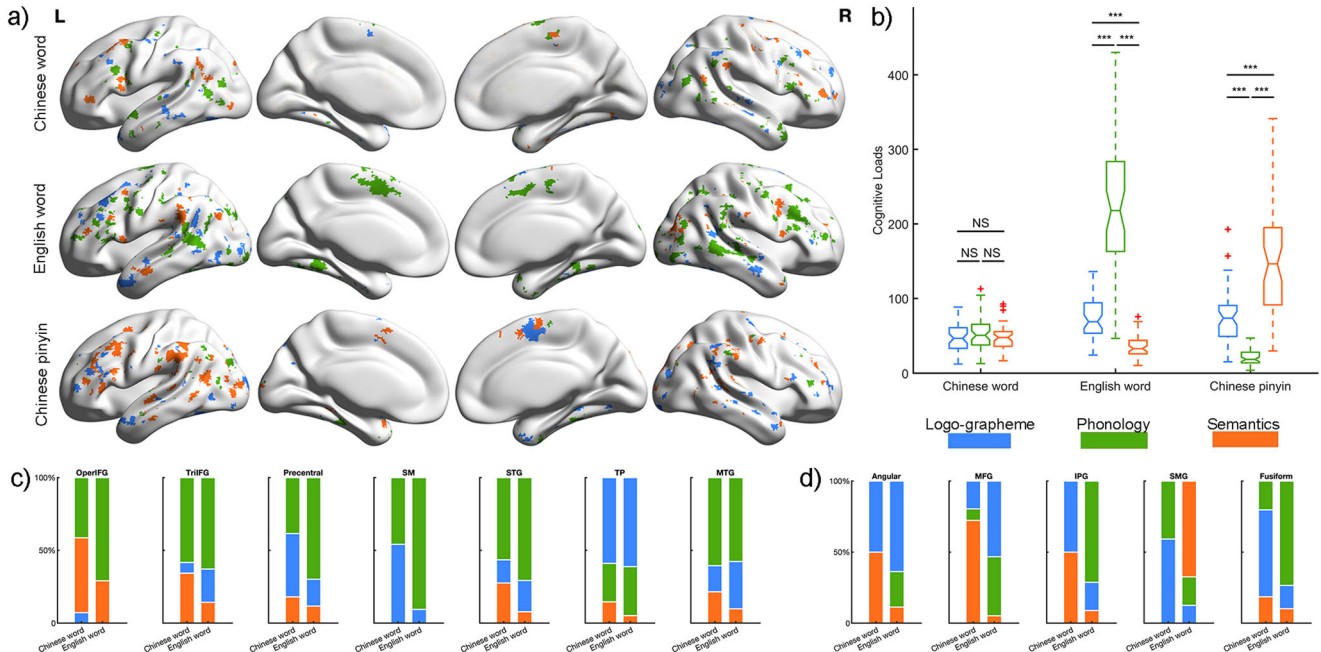

**Fig. 6 Brain representations for three linguistic components. a** Brain involvement maps of three linguistic components in each language type. Gray lines indicate the regional boundaries. **b** Two-way repeated ANOVA was performed, and significant differences were found between linguistic components and language types. Post hoc analysis shows differentiated patterns of brain activity elicited by language-processing components for English words and Chinese pinyin. \*\*\*$p < 0.001$; NS, non-significance. **c** Regions with similar patterns in percentage of cognitive loads for each linguistic component in Chinese word and English word reading. **d** Regions with different patterns in percentage of cognitive loads for each linguistic component in Chinese word and English word reading. Blue indicates brain activity for logo-grapheme, green represents brain activity for phonology and orange denotes brain activity for semantics.

## Discussion

In the current study, cortical encoding of three linguistic components (logo-grapheme, phonology, and semantics) in Chinese-English bilingual individuals was investigated across Chinese words, English words, and Chinese pinyin reading tasks. The three linguistic components were co-represented within most language-related cortical regions with separate neural populations. Specifically, distinct spatial distribution patterns detected for each type of brain representation across the three writing systems could classify the language category with 100% accuracy. Difference in linguistic component involvements were detected in the middle frontal gyrus, inferior parietal gyrus, angular gyrus, supramarginal gyrus, and fusiform gyrus in the left hemisphere. These differences might be due to the differences in linguistic properties, indicating the accommodation for the language-specific features. Meanwhile, the same co-representations in linguistic components were found in other core language-related regions, including the opercular parts of the inferior frontal gyrus, the triangular parts of the inferior frontal gyrus, temporal pole, superior and middle temporal gyrus, precentral gyrus and supplementary motor areas, indicating the assimilation of cognitive processing across languages. As a result of precision searchlight RSA, we showed both linguistic-feature-specific accommodation and assimilation between L1 and L2.

We found that language types could be correctly classified with brain activity corresponding to all components as well as to every specific component as features in the classification, which indicates that not only different linguistic components but also the same component across different language systems would entail distinguished neural populations which were adapted to their own linguistic features. Clinical findings also support this result, for brain damage may selectively impair the function of only one language in bilingual individuals; additionally, after therapeutic surgery or brain stimulation, recovery of only one language may occur in bilingual language-impaired patients[25–29]. Besides, our

finding about the greatest contribution of the left middle frontal areas in logo-grapheme representation for classification is meaningful for clinical applications, such as intraoperative localization. In addition, we found that most language-related regions are involved in two or three types of linguistic component processing. Previous findings on specific regions also support these findings. For example, both semantic and logo-grapheme representations were detected within the left fusiform[30]. Besides, employing intracranial high-density electrocorticography, Zhu et al. detected the distinct spatiotemporal patterns of syntactic and semantic processing within the human inferior frontal gyrus[31]. Recently, precision estimation of brain networks showed that distributed networks can fractionate into multiple specialized networks in association regions which may reflect evolutionary history[32,33]. An expansion-fractionation-specialization hypothesis was proposed, which is supported by the findings presented here.

For logo-grapheme processing, brain activity in response to Chinese word and English word reading showed shared regions in the right superior temporal gyrus (rSTG), left middle frontal gyrus (MFG), right fusiform gyrus (FG), and rIFG. One interesting point should be noted that spatially very close but not overlapping clusters were found in the rSTG across all of the pairs of reading systems (Chi-W ∩ Eng-W, rSTG, [60, −20, −10]; Chi-W ∩ Pin, rSTG, [70, −16, −8]; Pin ∩ Eng-W, rSTG, [66, −26, −6]). Previous studies demonstrated that the STG bilaterally is involved in visual-auditory integration both in English and Chinese reading[34–37]. Subtle anatomical separations supporting similar functions suggested that different visual-auditory linguistic features are integrated in subregions of the rSTG and this supports the accommodation hypothesis. Another interesting point is that brain loads of logo-grapheme of Chinese words were found relatively low in word form regions like FG and left occipital cortex but relatively high in high-order regions like left IFG, MTG. In consideration that logo-grapheme is the ideographic unit in Chinese as well as a graph-like component, brain loads of logo-grapheme indicate not only visual feature processing but also

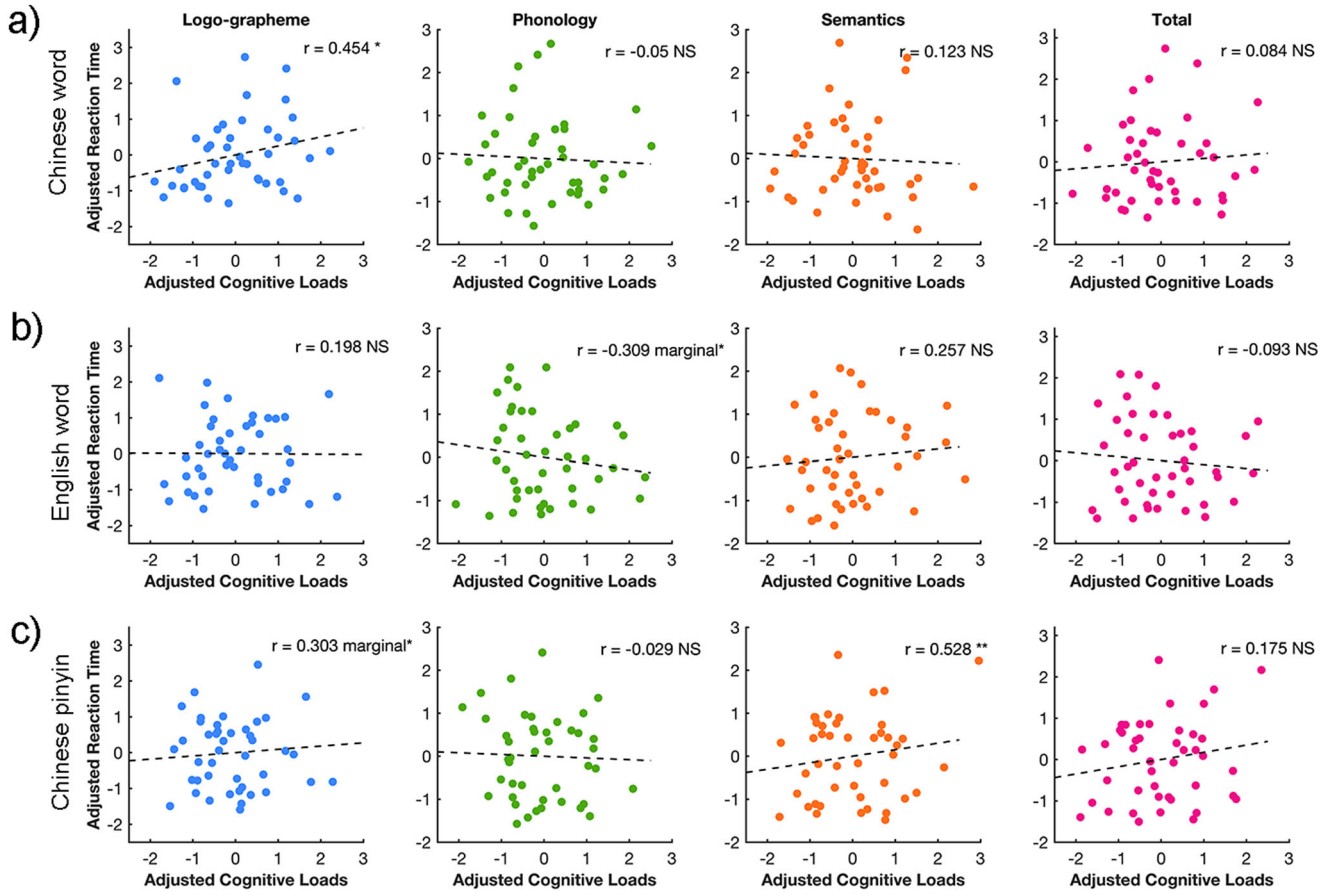

**Fig. 7 The association between brain cognitive loads and reaction times of every linguistic component in each language type.** Partial correlation was applied to calculate the relationship between brain activity elicited by language-processing components and reaction times for **a** Chinese words, **b** English words, and **c** Chinese pinyin.

orthographic processing. Our previous study reported logo-grapheme representations of real word and pseudo word (indicating visual feature processing and orthographic processing) were observed in fusiform gyrus and occipital cortices but logo-grapheme representations of false word (visual feature processing) were limited in occipital cortices, indicating a functional gradient in the fusiform cortex for Chinese word recognition[30]. A stereoelectroencephalography (SEEG) study reported directional connectivity from high-order brain regions (e.g., the left IFG, MFG, MTG) to word form regions (e.g., FG, ITG, MOG), suggesting top-down modulations occur in the early stage of Chinese word recognition[38]. Brain loads of logo-grapheme found in present study were consistent with previous studies and indirectly support the interactive view of cognitive processing in Chinese word recognition. Finally, the logo-grapheme brain loads in FG should be paid attention. The shared brain loads of Chinese word and English word located in the right fusiform gyrus might be considered a function of processing general features of objects or, more precisely, an adaptation of the second language, i.e., English, to the first language, i.e., Chinese. Notably, although previous studies considered the left ventral occipito-temporal cortex (vOT)/fusiform gyrus (known as visual word form area, VWFA) to be universally involved in visual word form processing, in the visual word form area, and this was script invariant in monolingual individuals[13,39], Gao and her colleagues found that subregions of the left ventral occipital-temporal cortex involved in visual word processing were different in Chinese-English bilingual individuals[40]. Also, it has been found that activation of the fusiform gyrus in English native speakers during reading is more left lateralized, but after learning Chinese, these individuals exhibited more bilateral fusiform activity during English reading[41]. In

this sense, the neural basis underlying Chinese as the first language would affect brain responses to reading in the second language[42].

For brain processing of phonology, although nonoverlap was found between the brain activity during reading Chinese word and Chinese pinyin, shared brain activity for logo-graphemes was found in the rSTG, rIFG, right inferior parietal gyrus, and right precentral gyrus, which are involved in visual-auditory integration[37,43]. Our results indicate that although phonological regions are involved, successful semantic access during Chinese reading depends more on morphological processing[44–47].

For the brain activations related to semantic processing, spatially close clusters were found in the left inferior frontal gyrus for Chi-W ∩ Pin (−38, 40, 6) and Pin ∩ Eng-W (−38, 42, 10), which is a brain region involved in semantic processing[48–51]. The overlap of brain activity between reading Chinese word and Chinese pinyin for semantic processing was also found in the left middle temporal gyrus and left angular gyrus, which played a core role in ideographic reading. In addition, an overlap of semantic processing during reading Chinese pinyin and English word was found in the left supramarginal gyrus, which was important for alphabetic-related reading.

In addition to the neuroanatomical separations for different linguistic processing, we found significant interactions between linguistic components and language types. Specifically, a relatively equal magnitude of brain activity pattern across the three linguistic components was observed for Chinese word reading. Given that cue effects of phonetic radicals on pronunciation are not obvious in Chinese word reading[52,53], the brain activity in response to phonological processing that was at a similar level to

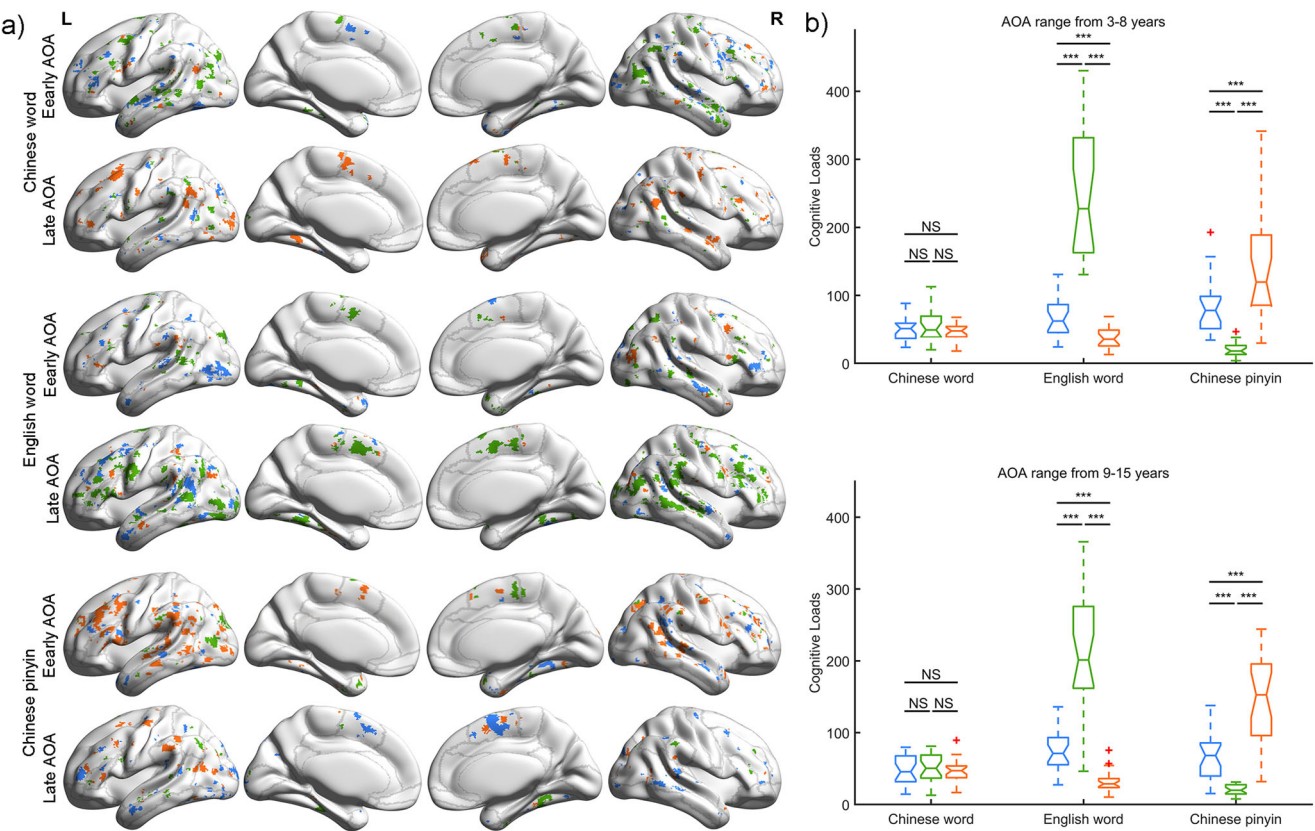

**Fig. 8 Brain loads for three linguistic components in early and late AOA participants. a** Brain involvement maps of three linguistic components in early and late AOA participants respectively among language types. Gray lines indicate the regional boundaries. **b** Two-way repeated ANOVA was performed, and significant differences were found between linguistic components and language types in early and late AOA participants respectively. Post hoc analysis shows differentiated patterns of brain activity elicited by language-processing components for English word and Chinese pinyin. ***$p < 0.001$; NS, non-significance.

logo-grapheme and semantics proved that phonological codes in Chinese word reading are activated obligatorily, in line with previous studies[46,54–56]. Interestingly, most activity related to phonological representation was found in reading English words and the least activity was related to reading Chinese pinyin. As for alphabetic writing systems, both English word and Chinese pinyin reading require readers to access an exact sound by sequentially mapping letters/letter combinations onto phonemes and assembling them together[44]. However, Chinese pinyin (corresponding to two-character words in the current study) can be mapped directly to two monosyllables with tones, while English words can be mapped onto polysyllables with 5 to 9 phonemes. Furthermore, regarding orthographic consistency, Chinese pinyin, as a phonetic coding system, is highly transparent, but English words are relatively opaque[57–61]. Thus, it is reasonable that during spelling, English words have a larger 'lexical competition cohort' and require more engagement for phonological processing, while Chinese pinyin completes grapheme-to-phoneme correspondence effortlessly[58]. It should be noted that the largest brain activity during semantic processing was found in Chinese pinyin reading, while the smallest was found in English word reading. For English words, once spelling was completed, exact sounds correspond to unique semantics, utilizing the grapheme-phonology-semantics pathway[62,63]. Given the richness of homophones in Chinese[64,65], more resources would be needed for phonology-to-semantics mapping during Chinese pinyin word reading. In addition, based on the direct visual-orthography to semantics hypothesis for Chinese characters, Chinese pinyin processing might include both the grapheme (Chinese pinyin)-

phonology-semantics pathway and the grapheme (Chinese pinyin)-phonology-logo-grapheme (Chinese word corresponded to Chinese pinyin)-semantics pathway[66,67].

Previous brain imaging studies utilizing traditional univariate analyses failed to link neural responses to detailed linguistic components and could only draw results at the cluster level. Our previous work used searchlight multivoxel pattern analysis to investigate voxel-wise neural response patterns of Chinese and English word reading in bilingual individuals[10]. Distributed neural populations in the left lateral occipital cortex, left fusiform gyrus, left temporal cortex, left temporal parietal cortex, left prefrontal cortex, and superior parietal cortex served for the dominant language and the second language respectively, which supports our results. On the other hand, with the development of deep learning and artificial intelligence, neural decoding techniques have achieved significant breakthroughs focused on single modalities, such as visual information[68] and articulatory movement[69]. However, there is still a large gap in decoding higher cognitive functions, such as language processing and decision making. Our findings shed light on the possibility of neural decoding for language comprehension. Notably, through mapping our representation maps to the HCPex template, we also found that the three language sub-networks calculated with effective connectivity in our previous work exhibit different involvements extent in each linguistic components[70].

AOA has been revealed as an important factor of affecting the distribution and amplitude of brain involvement for L2 in bilinguals[71,72]. Intersecting distributed neural population maps and similar brain regional representational patterns were found in

both early AOA and late AOA, which confirmed the view that divergent neural networks support convergent linguistic functions. Specifically, early AOA participants showed similar brain load maps for Chinese words and English words, indicating assimilation of L1 to L2. Also, late AOA participants utilized more extensive neural populations processing L2 than early AOA participants, which is consistent with previous studies[8] and highlights accommodation for L2 in late AOA bilinguals.

Two limitations of this study should be addressed. First, although searchlight RSA is a more fine-grained analysis method to show brain responses, it can only determine linguistic feature-related processes but fails to indicate what manipulation has occurred. New methods or combining searchlight RSA with subtle experimental designs should be developed in future to reveal more detail. Second, other factors, such as language proficiency have been revealed to influence brain responses during language processing, especially for bilinguals[73]. Future bilingual studies should take these points into consideration.

In summary, we found that the brain activity for the linguistic-processing components in different reading systems showed neuro-anatomically distinct spatial distributions. Brain activity patterns were significantly correlated with reaction time and could predict language types. Meanwhile, typical language-related regions had similar brain activation patterns and cognitive component processing of the three language types, Chinese words, English words, and Chinese pinyin. Taken together, both separated and shared brain activity patterns were associated with linguistic features of each language system. Our findings support and enrich the assimilation-accommodation hypothesis and demonstrate brain adaptation to long-term and complex language practice in bilingual individuals.

## Methods

**Participants.** Fifty-one participants (25 males; mean age = 23.4 years) enrolled in the current study through online advertising. All were Chinese native speakers, had vision/corrected vision over 4.8, and met the inclusion criterion of no history of neurological disease or psychiatric disorders. The subjects had started to learn English as their L2 between 3 and 15 years of age. According to age of acquisition (AOA), 23 subjects were early AOA (range from 3–8 years) and 28 subjects were late AOA (range from 9–15 years). All participants have received a unified education of Chinese pinyin at the elementary level in mainland China and passed the College English Test Band 4. All participants underwent an 8-min structural MRI scan and a task functional MRI scan that lasted approximately 40 min. The Edinburgh Handedness Inventory was used to identify participants' handedness[74]. Forty-one of all participants were classified as right-handed, and 10 participants had balanced dexterity. This study was approved by the Ethics Committee of the School of Life Sciences of Fudan University. Written informed consent was signed by every participant before the experiment.

**Stimuli and task-fMRI procedures.** In the current task-fMRI scans, an event-related design was utilized. Stimulus sets consisted of 3 conditions, Chinese word, Chinese pinyin, and English word, with 40 trials in each category. The Chinese word condition contained 40 two-character words; Chinese pinyin's corresponding Chinese words were also two-character words. All stimuli were white and presented on a black screen with a horizontal visual angle of 4.37°. To prevent confounding effects, picture size, percentage of pixels, number of strokes, and word frequency were matched across conditions. There was no semantic equivalent of any two stimuli across conditions. All 120 stimuli were visually presented for 1000 ms in a randomized order, and a fixation cross was presented for an interval of 4–6 s. The task was conducted in a single run with no stimuli repeated. Each stimulus subtended a visual angle of approximately 1° vertically and was presented in Song font for Chinese word and Arial for English words and Chinese pinyin in white against a black background. Participants were asked to respond to stimulus categories by pressing one of 3 different buttons with their index fingers as soon as possible once they recognized meanings of the stimulus. The matching between buttons and language categories was randomized across all participants. Notably, the current task did not require participants to respond at the fastest speed to avoid the situation in which participants directly press keys only according to visual features of Chinese Word, English Word, and Chinese Pinyin without semantic access. Thus, the accuracy rates and range of reaction time indicate whether the participants followed the experimental requirements and the time-length of reaction time is corresponding to cognitive recognition processing of the stimulus. The participants were told to complete a re-recognition questionnaire after scanning. The

participants needed to identify whether the words in the checklist were shown in the fMRI task. The checklist comprises 120 two-character Chinese words containing 60 never previously shown new words, 20 words from the Chinese word stimulus set, 20 words corresponding to the Chinese pinyin stimulus set, and 20 words semantically equivalent to the English word stimulus set. Given that memory performance was not the goal of the current study, the accuracy of the checklist was not used for further analysis. A practice experiment composed of 12 trials (4 stimuli for each condition) was performed before the normal fMRI scan to ensure that the participants fully understood the tasks. Because of a machine fault, only 44 participants' button pressing performance was successfully recorded and used for subsequent behavioral analysis.

**Image acquisition and data preprocessing.** Functional and anatomical images were acquired through a 3T Siemens Prisma scanner. An echo planar imaging (EPI) sequence was used for functional imaging data collection (echo time (TE) = 33 ms, flip angle = 52°, matrix size = 110*96, field of view = 220*196 mm, slice thickness = 2 mm, number of axial slices = 72, and repetition time (TR) = 720 ms). For anatomical reference, a high-resolution T1-weighted image was acquired before the tasks (TE = 2.56 ms, flip angle = 8°, matrix size = 320*320, field of view = 256*256 mm, slice thickness = 0.8 mm, number of sagittal slices = 208, and TR = 3000 ms).

Preprocessing and statistical analysis of fMRI data were performed using SPM12 (http://www.fil.ion.ucl.ac.uk/spm). For detail, slice timing was first conducted to obtain temporal realignment with the middle EPI volume. Unwrapped spatial realignment was performed to correct nonlinear distortions from head movement and magnetic field inhomogeneity. Next, the T1 image was coregistered to the mean EPI image. The coregistered image was segmented and normalized to Montreal Neurological Institute (MNI) space to obtain deformation field parameters. All realigned EPI volumes were spatially normalized to MNI space by applying the deformation field parameters. Finally, spatial smoothing with a 6 mm full-width half-maximum isotropic Gaussian kernel was performed for the normalized volumes.

**Activation analysis.** In the first-level analysis, a general linear model (GLM) was used for fixed-effect analysis of each participant for each condition. For the GLM model, the convolution of stimulus onset time and canonical hemodynamic response function served as independent variables; the blood oxygen level-dependent (BOLD) time series signals served as dependent variables with 6 realignment parameters as regressors. After high-pass filtering, contrasts of interest were obtained for each condition relative to fixation. In the second-level analysis, one-sample t-tests were performed to obtain an activation map of contrasts for each condition, FDR correction ($p < 0.05$), cluster size >10.

**Representational similarity analysis.** RSA has been widely used to evaluate congruent patterns across modalities within/between subjects[12,15–18]. Specifically, in the current study, RSA bridged brain activity patterns and behavioral measurements in response to task stimuli[75]. Brain activity patterns were depicted by constructing neural representational dissimilarity matrices (neural RDMs) that have $n*n$ dimensions ($n = 40$). Behavioral measurements were operationally defined by 3 behavioral representational dissimilarity matrices (logo-grapheme RDMs, phonological RDMs, and semantic RDMs), which are described in detail below.

*Behavioral RDMs.* To reveal the underlying internal neural mechanism for visual features and orthographic information processing, we constructed logo-grapheme RDMs by calculating one minus the overlapping ratio of basic units between any two stimuli in each language type[18]. For a Chinese word, the basic unit was a logo-grapheme that could not be semantically divided. Once a logo-grapheme was divided into some strokes, it would no longer carry semantic information. Notably, ideographic units and structural units are defined by different aspects, so one logo-grapheme might contain one or more strokes and be a part of radical or a radical itself. For example, the word "热情" (enthusiasm) is consists of 6 logo-graphemes (扌, 丸, 灬, 忄, 龶, 月); the word "眼睛" (eye) is composed of 5 logo-graphemes (目, 艮, 目, 龶, 月). They shared two logo-graphemes (龶, 月), so the dissimilarity is calculated as $1 - (2/(6 + 5)) = 0.818$. For Chinese pinyin, the basic logo-grapheme unit was a single "letter" or symbol for tone; and for English words, the basic unit was a single letter. Patterns for phonetic features were built by constructing phonological RDMs, calculated as one minus the ratio of shared phonetic units. The basic phonetic units were initials or finals or tones (regardless of position) for Chinese words and pinyin and vowels or consonants (regardless of position) for English words. For example, the Chinese pinyins of the word "热情" (enthusiasm) and word "眼睛" (eye) are composed of 6 units respectively (règíng, r, e, forth tone, q, ing, second tone; yǎnjīng, y, an, third tone, j, ing, first tone). They shared one unit (ing), so the dissimilarity is calculated as $1 - (1/(6 + 6)) = 0.917$. To construct the semantic RDMs, a skip-gram model for the word2vector algorithm (the software package was implemented in Python Gensim) was used to obtain continuous vector representations for each stimulus. The dissimilarity value between any two stimuli was calculated by one minus the cosine similarity between feature vectors corresponding to the stimuli. For Chinese word and pinyin, a wiki-Chinese corpus

was used as the input. Parameters were set as window size = 5, negative sample number = 5, dimension number = 300, learning rate = 0.025, and subsampling rate = 1e−4. For English words, same wiki-English corpus was used as input; parameters were set the same as they were for the Chinese word and Chinese pinyin. The correlations among logo-grapheme, phonological and semantic RDMs across three writing systems are shown in Supplementary Table 5.

*Neural RDMs and searchlight RSA.* A standard GLM for the first-level analysis was built to obtain trial-specific beta estimates for each participant. The first-level GLM contained 120 regressors corresponding to 40 stimuli in Chinese words, 40 stimuli in Chinese pinyin, and 40 stimuli in English words, with 6 head motion parameters regressed out as potential confounding factors. All regressors were convolved with the canonical hemodynamic response function (HRF) and high-pass filtered at 128 s.

After the first-level analysis, voxel-wise beta-value maps were obtained for 120 stimuli in each subject. In searchlight RSA, each voxel was extended into a self-centered spherical region of interest (ROI) with a 6-mm radius. One-level beta-values within each ROI for each condition were extracted and dissimilarity was calculated as one minus Spearman's rho between any two stimuli. After making the searchlight-center voxel pass through the cortex, we obtained 3 $n*n$ dimensional neural RDMs corresponding to 3 conditions in each voxel and each subject ($n = 51$). Here, to rule out the possibility that any functioning effects may be driven by non-linguistic types, a predefined language-related mask was employed. Specifically, the language-related anatomical mask includes 15 cortical regions that have been reported to be involved in language comprehension-related processing and their symmetric regions in the right hemisphere. More specifically, the mask was composed of the bilateral middle frontal (7#, 8#), inferior frontal (11#, 12#, 13#, 14#), precentral (1#, 2#), supplementary motor (19#, 20#), inferior parietal (61#, 62#), supramarginal (63#, 64#), angular (65#, 66#), superior temporal (81#, 82#), middle temporal (86#, 97#), superior temporal pole (83#, 84#), middle temporal pole (87#, 88#), fusiform (55#, 56#), superior occipital (49#, 50#), middle occipital (51#, 52#), inferior occipital (53#, 54#) areas through the Automated Anatomical Labeling (AAL)[76] template. In addition, a gray matter mask was used with a probability higher than 0.2 in the Tissue Probability Map (TPM) atlas in SPM12. After discarding voxels in which variations in the time series of BOLD signals were less than 1/8 of the mean values, the overlap of the gray matter mask and language mask was defined as the final mask used in the following analyses. For each component, the partial Spearman's rank correlation was calculated between neural RDMs in each voxel and one behavioral RDM (e.g., Logo-grapheme RDM) with two other behavioral RDMs (e.g., the phonology RDM and semantic RDM) controlled for each condition (Chinese word, English word, and Chinese pinyin). The whole-brain correlation maps for each linguistic component of each language in each subject were generated after the application of Fisher's r to z transformation to improve the normality.

We defined the cognitive loads as the sum of z values in voxels of which neural RDM was significantly positively correlated with the behavioral RDM, to represent the degree of brain activity for each linguistic component. For validation, brain activity was also calculated as the number of voxels significantly representing each linguistic component.

**Support vector regression.** To explore the association/dissociation of different cognitive brain activations derived from different writing system inputs, support vector regression (SVR) was applied (the software package was implemented in Python Scikit-learn, https://scikit-learn.org/stable/). For each subject, brain representations corresponding to 3 language-component representations in the 30 ROIs were extracted as feature vectors for each condition. On the other hand, categories of language-component representation were transformed into positive integers (Chinese word = 1, English word = 2, and Chinese pinyin = 3) as real label values. Thus, each participant contributed 3 feature samples with a size of 90 dimensions. A leave-one-subject-out (three samples contributed by each subject) cross-validation scheme was adopted to train the SVR model and test whether it could predict feature vector-related label values for languages. To avoid information leakage, for each fold (3 samples/subject), RSA results of the remaining 50 participants were used to generate group-level brain activation masks for logo-grapheme, phonology and semantics respectively. The training set and test set of brain activations were calculated based on a fold-specific group-level mask. Furthermore, a recursive feature elimination (RFE) scheme was adopted to reveal what features make the representation mechanism strikingly different across different writing systems. For standard evaluation, a unified group-level mask and leave-one-sample-out validation were adopted before conducting RFE. More specifically, each feature was eliminated, and the remaining features were fitted to the SVR model. The predictive contribution of a certain feature was measured by the root of the mean squared error (rMSE) of the SVR model fitted by the remaining 89 features. The higher the rMSE was, the worse the predictive performance was, and the more important were absent features. All features were sorted by absence-incurred rMSE in descending order, and absence-incurred rMSE smaller than the total rMSE ($N = 90$) were initially eliminated. Next, features ($N = 90$) were removed one by one in ascending order of absence-incurred rMSE, and then the remaining features ($N = 89, 88, 87…$) were used to fit the SVR model; then, the model was evaluated by negative rMSE. As a function of the remaining feature numbers $N$, the knee

point of negative rMSE revealed the best number of features for the most successful SVR model, and what those features are.

**Statistical analysis.** A two-way repeated ANOVA was conducted to test significant differences across conditions and linguistic components in reaction time, accuracy rate, and brain loads (all participants, early AOA participants, and late AOA participants respectively), with Bonferroni correction in post hoc tests. Spearman's rank correlation was utilized to depict the relationship among each RDM. A linear mixed-effects model was performed to determine an adjusted reaction time to exclude intra-subject effects. Then, partial correlation was conducted to reveal the relationship between the brain loads and fitted reaction time with Bonferroni correction. One-tailed one-sample t-tests ($H_0: \mu > 0$) were performed across participants to identify the voxels that were significantly involved in certain aspects of linguistic component processing for every condition at the group level. The threshold was set to $p < 0.05$, uncorrected, and cluster size >10.

**Validation analysis.** In avoiding the possibility that the separated anatomical patterns of cognitive component across languages were generated by chance, we conducted validation analysis to test replicability. Specifically, we randomly selected a subgroup with the number of participants ranging from 31 to 51 for 1000 times. The correlation values in cognitive loads between subgroup and the all participants were calculated. Besides, to remove the potential influence of handedness on our findings, we conducted the RSA analysis with only right-handed participants. The representation maps for each linguistic component with only right-handed subjects are presented, as well as the ones splitting the right-handed subjects into different groups of participants. In addition, to validate the illustration of our findings across different templates, the results were mapped to the HCPex template as well[24].

## Data availability
The original datasets supporting current study are not publicly available due to confidential personal information. The anonymous summary data and codes were available at https://github.com/ShujieGeng/Brain-Loads-for-Linguisitic-Components.

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

## Acknowledgements

We would like to thank Drs Yanchao Bi and Chu-Chung Huang for their insightful comments on the manuscript. This work was supported by the Natural Science Foundation of China [grant numbers 81901826, 61932008], the Natural Science Foundation of Shanghai [grant number 19ZR1405600, 20ZR1404900], the Shanghai Municipal Science and Technology Major Project [No. 2018SHZDZX01], the Scientific and Technological Innovation 2030 - the Major Project of the Brain Science and Brain-Inspired Intelligence Technology (No. 2021ZD0200500), ZJLab and Shanghai Center for Brain Science and Brain-inspired Technology. Professor Sir Colin Blakemore passed away on Jun 28th, 2022. We miss him so much and are deeply grateful for his substantial contribution to our research and insightful understanding on neuroscience.

## Author contributions

J.F., M.C., and C.B. contributed to the design of the work; S.G. and W.G. contributed to the acquisition of data, S.G. and M.C. contributed to the analysis of data; M.C., C.B., L.T., J.F., E.T.R., S.G., and W.G. contributed to the interpretation of data; S.G., M.C., and W.G. drafted the work; M.C., E.T.R., L.T., K.X., T.J., S.G., and W.Z. substantively revised the work.

## Competing interests

The authors declare no competing interests.
