## [Peer Review File · Communications Biology]

Reviewers' comments:

Reviewer #1 (Remarks to the Author):

This study aimed to reveal linguistic-feature specific accommodation and assimilation between first and second languages. Chinese-English bilinguals participated in an implicit reading task for Chinese words, English words and Chinese pinyin during fMRI scanning. The authors used RSA to examine the neural representations of three linguistic components (i.e., logo-grapheme, phonology, and semantics) of Chinese words, English words and Chinese pinyin, and found the convergent and divergent regions of each linguistic components among three types of languages. This research is novel in the bilingualism studies and the results are interesting. However, I have several major concerns on methodology and results.

Major concerns:

1. In this study, the authors used a relatively liberal threshold at the voxel level (i.e., $p < .05$, FDR corrected), which might run the risk of increasing type 1 errors.
2. As the fMRI task did not require participants to respond as fast as possible, it does not make sense to use reaction time as a dependent variable. In relation to this issue, it is not clear what the association between brain cognitive loads and reaction times reflects?
3. This study used a language categorization task for fMRI scan. But the authors thought differences in reaction time across the three conditions "used to indicate successful semantic access when pressing a button". Could the authors explain why RT differences in the language categorization task reflect the process of semantic access.
4. From the brain maps (Fig. 3A & 6A) and cognitive loads (Table S3) of three linguistic components for each language type, it seems that the network for each component is inconsistent with typical brain areas responsible for corresponding functions reported in previous studies. For example, the cognitive loads of logo-grapheme information of Chinese words was high in prefrontal cortex and temporal cortex, but low in the VWFA and left occipital cortex. The cognitive loads of phonological information of English words was high in the left MFG, but low in the temporoparietal region. The authors may need to explain this result.
5. In the reading task, three types of stimuli were used. There is possibility that the activations may related to language switching because the participants need to switch across languages. How do the authors control for the confounding effect of language switching?
6. The authors mentioned that the subjects had started to learn English as their L2 between 3 and 15 years of age. Due to the large span of AOA, it may be important to verify the results by controlling for the variance of AOA.

Minor points:

1. How many runs did the reading task include? Were the stimuli repeated during reading task? More detailed descriptions of the fMRI task are needed for the readers to understand the procedure.
2. I have several concerns about the construction of behavioral RDMs.
 - 1) For the logo-grapheme RDM of Chinese words, what is meant by "the basic unit was a logo-grapheme that could not be semantically divided"? It would be good if the authors gave examples of the construction of logo-grapheme, phonology, and semantics RDMs for three types of languages.
 - 2) For the phonological RDMs, the basic phonetic units were defined regardless of position. As the stimuli of Chinese word and Chinese pinyin were two-character words, the position of phonetic units should be taken into account.

Reviewer #2 (Remarks to the Author):

The authors have conducted a sophisticated brain imaging study to test the “assimilation-accommodation” hypothesis that the learning of a second language involves the use (assimilation) of language networks for the first language in addition to developing new neural networks to accommodate the special linguistic features of the second language. In this case, they studied brain activation patterns of English-Chinese bilingual speakers. They used a roving spotlight of RSA analysis to identify the relevant brain networks. The results support the assimilation-accommodation hypothesis, with some common regions involved in English, Chinese, and Pinyin word processing and other regions containing non-overlapping representations. I believe they will be of considerable interest to people working in this field. I have just a few questions/comments.

1. The Title refers to interdigitation of the network supporting the two different languages, but I think that description is a little misleading. Interdigitation means overlapping and/or alternating like the digits of two hands folded together, but the results seem to me to be more like multiple patches for different word forms present in the same general region of cortex. The arrangement of the patches seems more independent than interdigitated. I suggest using a different word.
2. The visual word form area is mentioned in the introduction, which cites some differences in results between single and dual language speakers. But then there is no further mention of the VWFA, at least that I can see. Are any of the networks identified here overlapping with the VWFA? What is their relationship to the classic VWFA in English speakers?
3. For temporal lobe, the areas that are involved in the processing of two or more types of characters, it was curious to see that they are mostly located dorsal to STS in the right hemisphere, but for the non-overlapping areas they are largely shifted ventral to STS. Is this correct? Any ideas for why this might be?
4. It is useful to see the results on the HCPex template in Figure S3, but it would be even more helpful to have a few of those template areas labeled, maybe in an accompanying figure?

Reviewers' comments:

Reviewer #1 (Remarks to the Author):

This study aimed to reveal linguistic-feature specific accommodation and assimilation between first and second languages. Chinese-English bilinguals participated in an implicit reading task for Chinese words, English words and Chinese pinyin during fMRI scanning. The authors used RSA to examine the neural representations of three linguistic components (i.e., logo-grapheme, phonology, and semantics) of Chinese words, English words and Chinese pinyin, and found the convergent and divergent regions of each linguistic components among three types of languages. This research is novel in the bilingualism studies and the results are interesting. However, I have several major concerns on methodology and results.

R: We would like to thank the editor and reviewers for all positive comments and offering the opportunity to revise our manuscript. We have addressed all comments and provided a point-to-point response as follows, with a denotation of the changes made to the manuscript accordingly.

Major concerns:

1. In this study, the authors used a relatively liberal threshold at the voxel level (i.e., $p < .05$, FDR corrected), which might run the risk of increasing type 1 errors.

R: We thank the reviewer for this comment. In the present study, activation analysis was performed to test the second-level fixed effects of Chinese words, English words and Chinese pinyin with a threshold: voxel-wise $p < 0.05$, FDR corrected $q < 0.05$, and cluster size > 10 (page 13 line 23).

For multiple comparison correction, Bonferroni correction is the conservative method in neuroimaging data analysis that hypothesizes voxels are independent with each other, and thus, the Type I error was controlled by setting the threshold as α/N (α is the significance level and N is the number of voxels). This assumption has shortcomings considering the fact that voxels in human brain are not purely independent with each other for both structural and functional constrains

In this sense, here we employed the False Discovery Rate (FDR) correction as an alternative to Bonferroni, which sets the proportion of incorrect rejections among all rejections of the null hypothesis at a tolerable level (i.e., usually as 0.05). Additionally, considering the spatial continuity, we controlled the cluster size (>10 voxels) as a compensation to the independent voxels hypothesis. This threshold is also widely used in neuroimaging studies for activation detection and group comparison.

To validate the results in the present study, we also showed the results with the threshold of Bonferroni correction and cluster size > 10 . The results with Bonferroni correction are the reduced versions of FDR correction ones, with all activation clusters and peak coordinates remained. Please see the left panel of the figure below for Bonferroni correction, with the right panel for FDR correction. Considering that

activation analysis is not the core analysis in the present study, we did not put this results in the revised manuscript.

2. As the fMRI task did not require participants to respond as fast as possible, it does not make sense to use reaction time as a dependent variable. In relation to this issue, it is not clear what the association between brain cognitive loads and reaction times reflects?

R: We thank the reviewer for this important comment. The ambiguity came from unclear writing which has been revised. Here, we aim to depict brain activity patterns of all linguistic components. In this sense, the instruction to participants is, after the semantic access of the stimuli, to press the correct button as fast as possible. We highlight that participants should not press the button as fast as possible to avoid the situation in which participants directly press keys ONLY according to visual features of Chinese Word, English Word and Chinese Pinyin without semantic access.

Therefore, the reaction time indicates the processing time length of word recognition. The association between brain cognitive loads and reaction time provides evidence that the higher the involvement of voxels for a linguistic component, the more relevant they are to word recognition. In this work, the mean reaction time for all participants is 968ms for Chinese words, 1182ms for English words and 1497ms for Chinese Pinyin, which are much longer than purely reaction to strokes (728ms) and passive viewing visual features (around 500ms) (Guo et al., 2022 eNeuro; Tian et al., 2020 Journal of Neurolinguistics), indicating the successes semantic processing.

We have paraphrased the relevant part of the Stimuli and Task-fMRI Procedures in the Methods section.

Page 12, lines 23-30: *‘Participants were asked to respond to stimulus categories by pressing one of 3 different buttons with their index fingers as soon as possible once they*

recognized meanings of the stimulus. The matching between buttons and language categories was randomized across all participants. Notably, the current task did not require participants to respond at the fastest speed to avoid the situation in which participants directly press keys only according to visual features of Chinese Word, English Word and Chinese Pinyin without semantic access. Thus, the accuracy rates and range of reaction time indicate whether the participants followed the experimental requirements and the time-length of reaction time is corresponding to cognitive recognition process of the stimulus.'

3. This study used a language categorization task for fMRI scan. But the authors thought differences in reaction time across the three conditions “used to indicate successful semantic access when pressing a button”. Could the authors explain why RT differences in the language categorization task reflect the process of semantic access?

R: We thank the reviewer for this comment. As stated above, we aim to explore the brain loads for different linguistic components during word recognition and asked participants to press the correct key only after they access semantics and as fast as possible. The differences in accuracy rates and range of reaction times indicate whether they followed the experimental requirements. Additionally, the reaction time would be much shorter and without group differences if participants responded based on visual features or orthography of stimuli instead of approaching semantics, indicating the successes semantic processing.

We have paraphrased the relevant sentences. More details are provided in the *Stimuli and Task-fMRI Procedures* in the *Methods* section.

Page 12, lines 23-30: *‘Participants were asked to respond to stimulus categories by pressing one of 3 different buttons with their index fingers as soon as possible once they recognized meanings of the stimulus. The matching between buttons and language categories was randomized across all participants. Notably, the current task did not require participants to respond at the fastest speed to avoid the situation in which participants directly press keys only according to visual features of Chinese Word, English Word and Chinese Pinyin without semantic access. Thus, the accuracy rates and range of reaction time indicate whether the participants followed the experimental requirements and the time-length of reaction time is corresponding to cognitive recognition process of the stimulus.’*

4. From the brain maps (Fig. 3A & 6A) and cognitive loads (Table S3) of three linguistic components for each language type, it seems that the network for each component is inconsistent with typical brain areas responsible for corresponding functions reported in previous studies. For example, the cognitive loads of logographeme information of Chinese words were high in prefrontal cortex and temporal cortex, but low in the VWFA and left occipital cortex. The cognitive loads of phonological information of English words were high in the left MFG, but low in the temporoparietal region. The authors may need to explain this result.

R: We thank the reviewer for this insightful comment. This work focused on the bilingual comparisons to test the assimilation-accommodation hypothesis, so we only have little discussion about monolingual brain load results in the previous draft. We have added further discussion about these issues now.

For question 1 *‘the cognitive loads of logo-grapheme information of Chinese words were high in prefrontal cortex and temporal cortex, but low in the VWFA and left occipital cortex’*:

Specifically, brain loads for logo-grapheme in the VWFA in bilinguals were discussed in the third paragraph in the *Discussion* section. In monolinguals, VWFA was involved in word recognition by functional gradient activation pattern and interacted with high-order regions in the early stage, which support the interaction view of brain in processing script. In the revised draft, we added the comparisons with previous studies and explanations listed as below and added to *the third and fourth paragraphs in the Discussion section* of the revised draft.

Page 9, lines 12-23: *“Another interesting point is that brain loads of logo-grapheme of Chinese word were found relatively low in word form regions like FG and left occipital cortex but relatively high in high-order regions like left IFG, MTG. Considering that logo-grapheme is the ideographic unit in Chinese as well as a graph-like component, brain loads of logo-grapheme indicate not only visual feature processing but also orthographic processing. Our previous study reported that logo-grapheme representations of real words and pseudowords (indicating visual feature processing and orthographic processing) were observed in the fusiform gyrus and occipital cortices, but logo-grapheme representations of false words (visual feature processing) were limited in occipital cortices, indicating a fusiform functional gradient for Chinese word recognition (Guo et al., 2022). A stereoelectroencephalography (SEEG) study reported directional connectivity from high-order brain regions (e.g., the left IFG, MFG, MTG) to word form regions (e.g., FG, ITG, MOG), suggesting that top-down modulations occur in the early stage of Chinese word recognition (Y. Liu et al., 2021). The brain loads of logo-grapheme found in the present study were consistent with previous studies and indirectly support the interactive view of cognitive processing in Chinese word recognition.”*

For question 2: *‘The cognitive loads of phonological information of English words were high in the left MFG, but low in the temporoparietal region.’*

As shown in Table S3, the brain loads of phonological information of English words were relatively high in the left MFG (7.91), IPG (8.94), and STG (7.86) and also high in the MTG (16.89) and precentral gyrus (13.84).

5. In the reading task, three types of stimuli were used. There is possibility that the activations may related to language switching because the participants need to switch across languages. How do the authors control for the confounding effect of language switching?

R: We thank the reviewer for this comment. We totally agree with the reviewer that

language switching effects are an important confounding factor when more than one language stimulus are presented. In this work, we tried to eliminate the language switching effects through three aspects. First, the time length of the stimulus interval was set as 4-6 s, which theoretically allowed the haemodynamic response to reach the peak point. In other words, the next stimulus was presented after the complete haemodynamic peak point response of the previous stimulus. Second, 120 stimuli (40 Chinese words, 40 English words and 40 Chinese pinyin) were randomly presented to counterbalance possible language switching effects and habit effects. Finally, representational similarity analysis (RSA) was utilized to identify the voxels associated with a specific linguistic component instead of a contrast between two conditions. It statistically minimized the effects of other cognitive processes, including language switching. Based on the above three points, language switching effects were thought to be controlled and not taken into consideration as confounding variables in subsequent analysis.

6. The authors mentioned that the subjects had started to learn English as their L2 between 3 and 15 years of age. Due to the large span of AOA, it may be important to verify the results by controlling for the variance of AOA.

R: We thank the reviewer for this very important comment. Age of second language acquisition plays an important role of effecting neural populations underlying L2 and L1 processing in bilinguals. Given that how AOA effects the distribution and processing of neural networks for L2 and L1 in bilinguals is sophisticated and controversial, controlling effects of AOA by linear regression might be inappropriate. Thus, we divided participants into early AOA (3-8 years, 23 subjects) and late AOA (9-15 years, 28 subjects) to detect the fine-grained neural networks supporting linguistic components for different language types.

Brain loads of 3 linguistic components for Chinese word, English word and Chinese pinyin were repeated within each AOA group. We found that early AOA subjects showed similar brain loads pattern for L2 and L1 while late AOA subjects utilized more extensive neural populations processing L2 than L1 ($P_{\text{permutation}} = 0.060$, 10,000 times), which is consistent with previous studies and support Assimilation-Accommodation hypothesis. Despite of AOA, significant interactions between language types and linguistic components were remain: no-significant differences of brain loads among linguistic components in Chinese word, reversed V shape in English word and V shape in Chinese pinyin, denoting brain adaptations for languages. We updated the relevant sections in the revised draft and listed below:

Page 11, line 7: *‘According to age of acquisition (AOA), 23 subjects were early AOA (range from 3-8 years) and 28 subjects were late AOA (range from 9-15years)’.*

Page 16, line 14: *‘A two-way repeated ANOVA was conducted to test significant differences across conditions and linguistic components in reaction time, accuracy rate, and brain loads (all participants, early AOA participants and late AOA participants respectively)’.*

Page 8, line 27: *'Given that age of acquisition (AOA) for second language has potential effects on the distribution and involvement of neural networks supporting L2 and L1 in bilinguals, we divided participants into early AOA subgroup (3-8 years) and late AOA subgroup (9-15 years) and depicted brain load maps of linguistic components for early AOA and late AOA respectively. As shown in Figure 8A, both early AOA and late AOA participants showed intersecting neural populations underlying linguistic components for Chinese word, English word and Chinese pinyin. Specifically, early AOA participants showed similar brain load maps of linguistic components between Chinese word and English word. Compared with early AOA participants, late AOA participants activated more extensive brain regions processing L2 ($P_{\text{permutation}} = 0.060$, 10,000 times). Furthermore, at regional level, despite of AOA, significant interactions between language types and linguistic components were remain: no-significant differences of brain loads among linguistic components in Chinese word, reversed V shape in English word and V shape in Chinese pinyin, denoting brain adaptations for languages.'*

Page 11, line 31: *'AOA has been revealed as an important factor of effecting distribution and amplitude of brain involvement for L2 in bilinguals (Mayberry, Chen, Witcher, & Klein, 2011; Wei et al., 2015). Intersecting distributed neural population maps and similar brain regional representational patterns were found in both early AOA and late AOA, which confirmed the view that divergent neural networks supporting convergent linguistic functions. Specifically, early AOA participants showed similar brain loads maps for Chinese word and English word, indicating assimilation of L1 to L2. Also, late AOA participants utilized more extensive neural populations processing L2 than early AOA participants, which is consistent with previous studies (H. Liu & Cao, 2016) and highlights accommodation for L2 in late AOA bilinguals.'*

Page 30, line 1:

*"Figure 8. Brain loads for three linguistic components in early and late AOA participants. A. Brain involvement maps of three linguistic components in early and late AOA participants respectively among language types. Grey lines indicate the regional boundaries of AAL. B. Two-way repeated ANOVA was performed, and significant differences were found between linguistic components and language types in early and late AOA participants respectively. Post hoc analysis shows differentiated patterns of brain activity elicited by language-processing components for English word and Chinese pinyin. ***: $p < 0.001$."*

Minor points:

1. How many runs did the reading task include? Were the stimuli repeated during reading task? More detailed descriptions of the fMRI task are needed for the readers to understand the procedure.

R: The task was conducted in a single run with no repeated stimuli. In consideration of the limit experimental time length and possible fatigue effect, we have to make a trade-off between the number of stimuli and the number of repetitions. The RSA searched associations between neural RDM and behavioural RDM (both were $N \times N$ matrices, N indicates the number of stimuli in one language category, in the current study, $N = 40$). Few numbers of stimuli would reduce the reliability of RSA results, so we set N as 40. Repetition of stimuli would multiple total time-lengths, and few repetitions would not significantly increase the reliability of the first-level results. Therefore, we chose to conduct the task in a single run with no repeated stimuli. We have added relevant descriptions to the *Stimuli and Task-fMRI Procedures of Methods section*.

Page 12, line 21: *'The task was conducted in a single run with no stimuli repeated.'*

2. I have several concerns about the construction of behavioural RDMs.
 - 1) For the logo-grapheme RDM of Chinese words, what is meant by "the basic unit was a logo-grapheme that could not be semantically divided"? It would be good if the authors gave examples of the construction of logo-grapheme, phonology, and semantics RDMs for three types of languages.
 - 2) For the phonological DMs, the basic phonetic units were defined regardless of position. As the stimuli of Chinese word and Chinese pinyin were two-character words, the position of phonetic units should be taken into account.

R: We thank the reviewer for this comment. The elements of logo-grapheme RDM and

phonetic RDM were both calculated as follow: 1 minus the shared ratio of two stimuli.

Chinese words are made of characters. Characters are made of one or more radicals. Radicals may suggest phonetic or semantic information. Stroke is the basic structural unit of Chinese characters. These three elements are categorized by visually structural features. Considering that Chinese is ideographic and orthographic processing is more important than visual feature processing, we used an ideographic unit, logo-grapheme, to calculate RDM. “The basic unit was a logo-grapheme that could not be semantically divided” means that once a logo-grapheme was divided into some strokes, it would no longer carry semantic information. Notably, ideographic units and structural units are defined by different aspects, so one logo-grapheme might contain one or more strokes and be a part of radical or a radical itself. A character was divided into logo-graphemes based on the *Chinese Character Component Standard of GB 13000.1 Character Set for Information Processing*.

We added the relevant explanation and examples of logo-grapheme and phonology dissimilarity calculation in the *Behavioural RDMs/Representational Similarity Analysis/Methods* section.

Page 14, line 4: “Once a logo-grapheme was divided into some strokes, it would no longer carry semantic information. Notably, ideographic units and structural units are defined by different aspects, so one logo-grapheme might contain one or more strokes and be a part of radical or a radical itself. For example, the word “热情”(enthusiasm) is consist of 6 logo-graphemes(扌 , 丸 , 灬 , 丿 , 𠃉 , 月); the word “眼睛”(eye) is composed of 5 logo-graphemes(目 , 艮 , 目 , 𠃉 , 月). They shared two logo-graphemes (𠃉 , 月), so the dissimilarity is calculated as $1 - (2 / (6 + 5)) = 0.818$.”

Page 14 line11: For example, the Chinese pinyins of the word “热情”(enthusiasm) and word “眼睛”(eye) are composed of 6 units respectively (rèqíng, r, e, forth tone, q, ing, second tone; yǎnjīng, y, an, third tone, j, ing, first tone). They shared one unit (ing), so the dissimilarity is calculated as $1 - (1 / (6 + 6)) = 0.917$. ”

For phonological RDM, it is calculated regardless of position mainly due to the following considerations. The BOLD signal of task fMRI is the indirect reflection of neuronal activities and is ideally represented as the convolution of task stimulus and haemodynamic response function. The sampling frequency of the BOLD signal was one TR (TR = 720 ms in the present study), while a two-character word (3-9 phonemes) was routinely pronounced within approximately 1000 ms. Thus, the BOLD signal itself contains less information on subtle differentiated phonological processing. Furthermore, although phonological process-related ROIs were activated during word recognition, auditory stimuli were not presented. Generally, internal “reading” is not as rigorously

pronounced as real articulation. Above all, the position of phonetic units was not taken into account when calculating RDM.

For semantic RDM calculation, it is based on the word2vector algorithm, through which semantics of each stimulus would be represented by a 300-dimensional vector. The element of semantic RDM was calculated as follows: 1 minus the cosine angle between the two 300-dimensional vectors. The semantic vectors of all stimuli and all RDMs are available on GitHub (<https://github.com/ShujieGeng>).

Reviewer #2 (Remarks to the Author):

The authors have conducted a sophisticated brain imaging study to test the “assimilation-accommodation” hypothesis that the learning of a second language involves the use (assimilation) of language networks for the first language in addition to developing new neural networks to accommodate the special linguistic features of the second language. In this case, they studied brain activation patterns of English-Chinese bilingual speakers. They used a roving spotlight of RSA analysis to identify the relevant brain networks. The results support the assimilation-accommodation hypothesis, with some common regions involved in English, Chinese, and Pinyin word processing and other regions containing non-overlapping representations. I believe they will be of considerable interest to people working in this field. I have just a few questions/comments.

R: We thank the reviewer for all positive and insightful comments. We have addressed all comments and provided a point-to-point response as follows, with a denotation of the changes made to the manuscript accordingly.

1. The Title refers to interdigitation of the network supporting the two different languages, but I think that description is a little misleading. Interdigitation means overlapping and/or alternating like the digits of two hands folded together, but the results seem to me to be more like multiple patches for different word forms present in the same general region of cortex. The arrangement of the patches seems more independent than interdigitated. I suggest using a different word.

R: We thank the reviewer for this valuable suggestion and have revised the title as “Intersecting distributed networks support convergent linguistic functioning across different languages in bilinguals”. As the reviewer mentioned, brain networks underlying 3 linguistic components for different language types depicted in the current study seem more independent than interdigitated, but the word ‘interdigitated’ highlighted adjacent and/or similar. After deliberation, we use ‘intersecting’ to replace ‘interdigitated’, as ‘intersecting’ indicates that different brain networks meet at some junctions but are mainly divergent.

2. The visual word form area is mentioned in the introduction, which cites some differences in results between single and dual language speakers. But then there is no further mention of the VWFA, at least that I can see. Are any of the networks

identified here overlapping with the VWFA? What is their relationship to the classic VWFA in English speakers?

R: We thank the reviewer for this important comment. The relevant discussion can be found in the revised manuscript.

Page 9, line 27: *“Notably, although previous studies considered the left ventral occipito-temporal cortex (vOT)/fusiform gyrus (known as visual word form area, VWFA) to be universally involved in visual word form processing and script invariant in monolingual individuals (Krafnick et al., 2016; Price, 2012), Gao and her colleagues found that subregions of the left vOT were involved in different visual word processing in Chinese-English bilingual individuals (Gao et al., 2017). Additionally, it has been found that activation of the fusiform gyrus in English native speakers during reading is more left lateralized, but after learning Chinese, these individuals exhibited more bilateral fusiform activity during English reading (Mei et al., 2015). In the present study, the FG of Chinese-English bilinguals showed right-lateralized logo-grapheme brain loads for both Chinese words and English words (Table S3), which supported the view that the neural basis underlying Chinese as the first language would affect brain responses to reading in the second language (Tan, 2005).”*

3. For temporal lobe, the areas that are involved in the processing of two or more types of characters, it was curious to see that they are mostly located dorsal to STS in the right hemisphere, but for the non-overlapping areas they are largely shifted ventral to STS. Is this correct? Any ideas for why this might be?

R: We thank the reviewer for this comment. For brain loads of logo-grapheme, overlaps between any two language types were found in the right STG. We have added the relevant discussions.

Page 9, Line 5: *“One interesting point should be noted that spatially very close but not overlapping clusters were found in the rSTG across all of the pairs of reading systems ($Chi-W \cap Eng-W$, rSTG, [60, -20, -10]; $Chi-W \cap Pin$, rSTG, [70, -16, -8]; $Pin \cap Eng-W$, rSTG, [66, -26, -6]). Previous studies have demonstrated that the STG bilaterally is involved in visual-auditory integration both in English and Chinese reading (Holloway, van Atteveldt, Blomert, & Ansari, 2015; Kast, Bezzola, Jancke, & Meyer, 2011; McNorgan, Randazzo-Wagner, & Booth, 2013; Xia, 2020). Subtle anatomical separations supporting similar functions suggested that different visual-auditory linguistic features are integrated into subregions of the rSTG, and this supports the accommodation hypothesis”.*

Non-overlapping brain loads were distributed in both the right STG and STG, as shown in Fig. 4, Fig. 6 and Table S3. The brain loads with linguistic components were identified in the current study. In future studies, the detailed cognitive manipulation underlying brain load should be investigated.

4. It is useful to see the results on the HCPex template in Figure S3, but it would be even more helpful to have a few of those template areas labeled, maybe in an accompanying figure?

R: We thank the reviewer for this comment. Figure S3 has been updated with illustrations of HCPex atlas on the right panel.

REVIEWERS' COMMENTS:

Reviewer #1 (Remarks to the Author):

The authors have addressed all of my concerns.

Reviewer #2 (Remarks to the Author):

My questions and concerns have been addressed.